# Pigment Production by *Pseudofusicoccum* sp.: Extract Production, Cytotoxicity Activity, and Diketopiperazines Identified

**DOI:** 10.3390/microorganisms13020277

**Published:** 2025-01-26

**Authors:** Bianca Vilas Boas Alves, Letícia Jambeiro Borges, Samira Abdallah Hanna, Milena Botelho Pereira Soares, Daniel Pereira Bezerra, Laysa Lanes Pereira Ferreira Moreira, Warley de Souza Borges, Ricardo Wagner Dias Portela, Clara Couto Fernandez, Marcelo Andrés Umsza-Guez

**Affiliations:** 1Food Science Postgraduate Program, Faculty of Pharmacy, Federal University of Bahia, Salvador 40170-100, BA, Brazil; biancavbalves@gmail.com (B.V.B.A.); leticiajambeiro@gmail.com (L.J.B.); 2Laboratory of Applied Microbiology of the Health Sciences Institute, Federal University of Bahia, Salvador 40110-100, BA, Brazil; samirah@ufba.br; 3Gonçalo Moniz Institute, Oswaldo Cruz Foundation (IGM-FIOCRUZ/BA), Salvador 40296-710, BA, Brazil; milena.soares@fiocruz.br (M.B.P.S.); daniel.bezerra@bahia.fiocruz.br (D.P.B.); 4SENAI Institute for Innovation in Advanced Health Systems, SENAI CIMATEC, Salvador 41650-010, BA, Brazil; 5Chemistry Postgraduate Program, Center for Exact Sciences, Federal University of Espírito Santo, Vitória 29075-910, ES, Brazil; laysalannes@hotmail.com (L.L.P.F.M.); warley.borges@ufes.br (W.d.S.B.); 6Biotechnology Department, Federal University of Bahia, Salvador 40110-902, BA, Brazil; rwportela@ufba.br (R.W.D.P.); claracfz@gmail.com (C.C.F.)

**Keywords:** food industry, endophytic fungi, natural colorant

## Abstract

Filamentous fungi are among the most commonly used microorganisms for producing various metabolites including dyes. Ensuring the safety of products derived from microorganisms is always essential. In this study, the isolated fungus was identified as *Pseudofusicoccum* sp., a producer of the burgundy pigment through submerged fermentation. The fungus exhibited enhanced growth and pigment production under yellow light. The extract obtained showed no cytotoxicity in the tested cell lines (HepG2, SCC4, BJ, and MRC-5). Among the compounds isolated and identified through NMR analysis, cyclo(L-Pro-L-Val) and cyclo(L-Leu-L-Pro) (diketopiperazines) had been previously reported in foods and are known to be produced by various organisms, with several beneficial biological activities. This identified fungus represents a promising source of biopigments with a crude extract that is non-cytotoxic. Additionally, the isolated compounds exhibit significant biological properties, such as antibacterial, antifungal, and antioxidant activities, highlighting their potential as natural pigments for use in food products.

## 1. Introduction

Data on strategies for discovering new fungal strains and optimizing their industrial-scale production remain limited. Biotechnological processes emphasize the isolation of hyperproducing strains (endophytic fungi) for biopigment production [1,2,3]. Fungi are significant producers of secondary metabolites with noteworthy properties, including biocontrol agents, antimicrobials, antitumor agents, antioxidants, antidiabetics, antibiotics, insecticides, and pigments among others [4,5].

Filamentous fungi are among the most commonly used microorganisms for dye production [6,7]. Examples include *Monascus*, *Aspergillus*, *Penicillium*, *Fusarium*, *Trichoderma*, *Talaromyces*, *Neoscytalidium*, and *Diaporthe* among many others that are described as producers of biopigments [8,9] such as red [10], yellow [11], orange [12], and pink/violet [13]. These biopigments (melanins, phenazines, flavins, carotenoids, quinones, violacein, indigo, monascins, rubropunctamine, rubropunctatin, and ankaflavin) usually exhibit antimicrobial and antioxidant activities, attracting the attention of the dye industry due to their low or negligible toxicity [14,15].

Fungi also produce biocompounds beyond pigments such as diketopiperazines. These compounds, extensively documented in the literature, have been identified in a wide range of natural products, as well as in processed foods including beverages and ingredients [16]. Some species of microorganisms produce these compounds, which are associated with antimicrobial (antibacterial, antifungal, and antiviral), antioxidant, immunoregulatory, and other biological activities [17]. Their presence in foods offers several advantages.

The global food color market was valued at USD 4088.33 million in 2021 and is expected to expand, reaching USD 6206.56 million by 2027 [18]. Food-grade pigments require approval from regulatory agencies, with toxicological safety being a critical prerequisite [19,20]. The application of fungal biopigments in products, such as in foods, depends on factors including molecular stability, the absence of toxicogenic compounds, and economic feasibility (e.g., production yield) [21].

Despite their long history of use, microbial pigments are banned in some countries due to the potential presence of toxic secondary metabolites [22]. This restriction imposes commercial limitations, as seen with *Monascus* pigments, traditionally used as food coloring in Asia but banned in Europe and the United States [22,23].

One significant commercial limitation of fungal pigments is the toxic nature of certain fungal metabolites [22]. Therefore, studies are essential to evaluate the toxicity potential, confirm the non-mycotoxigenic nature of the fungi used, and identify the chemical structures of associated metabolites [24]. Some cytotoxicity studies with endophytic fungi have demonstrated cytotoxic activity against certain cancer cell lines [25,26].

Pigment biosynthesis is heavily influenced by fermentation conditions such as medium composition and process parameters. In most organisms, light, like temperature, is a crucial environmental signal regulating developmental and physiological processes [27,28]. These responses are mediated by photoreceptors that initiate signal transduction, leading to changes in gene expression encoding enzymes responsible for mycelial growth and secondary metabolite production in fungi [29].

The effects of light have been investigated in model fungal species, including in morphological studies on *Coprinus* (a basidiomycete) and *Phycomyces* (a zygomycete) [30], molecular studies on *Neurospora crassa* (an ascomycete) [31], and studies on reproductive structures in *A. nidulans* [32]. However, little is known about how different light colors affect fungal growth and pigment production in synthetic media.

Considering that fungi produce various types of metabolites, particularly under stress conditions caused by process variables, this study aimed to identify the isolated fungus, evaluate the influence of the light type on pigment production, analyze cytotoxicity, and identify some compounds present in the obtained extract. The main contribution of this research is to provide scientific evidence of the extract’s potential and safety, focusing on cytotoxicity assessment and its potential future applications.

## 2. Materials and Methods

### 2.1. Fungus Cultivation

A fungus was isolated from *Manilkara salzmannii* leaves located in Parque das Dunas in Salvador, Bahia, Brazil (12°58′16″ S and 38°30′39″ W). The fungal isolate was cultured in Petri plates containing Potato Dextrose Agar (PDA) at 28 ± 2 °C (301.15 K ± 275.15 K) and sub-cultured once every week according to Petrini et al. [33]. During the cultivation of the isolated fungus in solid and liquid media, it was observed that after a few days, the culture medium acquired a red color. This fact sparked interest in identifying the fungus and the characteristics of the pigment extract produced.

### 2.2. Fungal Identification

A molecular biology analysis was conducted for fungal identification, as suggested by Fonseca et al. [34]. Briefly, genomic DNA samples from the isolated fungus were obtained and purified using commercially available kits (Thermo Scientific, Waltham, MA, USA) and polymerase chain reactions (PCRs) were performed using the LROR/LR7 and ITS4/ITS5 primers for the amplification of the large subunit (LSU) rDNA and of the complete internal transcribed region, respectively. Subsequently, the amplicons were sequenced through the Sanger method (ABI PRISM 3130xl, Applied Biosystems, Foster City, CA, USA) using the BigDye Terminator v3.1 Cycle Sequencing Kit (Applied Biosystems, Waltham, MA, USA). The acquired concatenated consensus sequences were submitted to the Basic Local Alignment Search Tool (BLAST) for comparison and identification by identity analysis with the nucleotide sequence database (nr) from the GenBank of the National Center for Biotechnology Information (NCBI).

### 2.3. Submerged Fermentation and Pigment Extract Production

The fungus was incubated for seven days in PDA with the objective to be used as an inoculum for submerged fermentation. This methodology was conducted according to Abiala, Ogunjobi, Odebode and Ayodele [35] with some modifications. Three mycelial agar discs (5 mm diameter) were transferred to 100 mL of sterile Potato Dextrose Broth (PDB), pH 8.5, in Erlenmeyer Flasks. To study the effects of different wavelengths of light on growth and pigment production, the experiment was set up based on the principle that a colored glass paper allows only its color of light to pass through—it filters out the other colors of the spectrum (Figure 1). The flasks were covered in colored glass papers of blue (492–455 nm), green (577–492 nm), yellow (597–577 nm), or red (780–622 nm) and placed under an 18 W light source (LED cold light, 1350 lumens—with a light source centralized) inside an incubator and kept at 28 ± 2 °C (301.15 K ± 275.15 K) for 21 days. Negative controls were covered with aluminum foil (distributed in such a way as to ensure that all flasks received the same amount of light) at 28 ± 2 °C for 21 days (Figure 2). After the incubation, the supernatant was filtered using 0.45 µm filters coupled to a vacuum pump. The supernatants were stored in a flask covered with aluminum foil at 4 °C (277.15 K) for further analysis.

### 2.4. Effect of Light on Pigment Production

A PDB sample, without fungal inoculum, was kept as blank for pigment analysis so that any colored substances from the solid substrate were subtracted from the pigment produced by the fungus. The pigment analysis was carried out in a spectrophotometer (Multiskan FC—Thermo Scientific). The samples of pigment extracts (produced with different colors light and absence of light) were read at 420, 520, and 620 nm (wavelengths that represent the maximum absorption for yellow, reddish-brown, and blue pigments, respectively (Figure 1 and Figure 2)), taking into consideration the dilution factor of the sample [36]. All analyses were conducted in duplicate. Pigment yield was expressed in optical density units per gram of dry fermented matter multiplied by its dilution factor [37]. The absorbance values were converted to color units.

### 2.5. Biomass Estimation

The fungal biomass obtained during filtration of the submerged fermentation was washed twice with deionized water, followed by drying at 105 °C (378,15 K) for 12–15 h and biomass weighting. This methodology was conducted according to Velmurugan et al. (2010) [31].

### 2.6. Cell Cultures

The cytotoxicity assay was performed with HepG2 (human hepatocellular carcinoma), SCC4 (oral human squamous carcinoma), BJ (human foreskin fibroblasts), and MRC-5 (human lung fibroblast) cells, all of them obtained from the American Type Culture Collection (ATCC). Cells were grown in culture bottles (75 cm^3^, 250 mL) using RPMI 1640 medium supplemented with 10% fetal bovine serum and kept in incubators with 5% CO_2_ atmosphere at 37 °C (31,015 K). Cell growth was monitored daily using an inverted microscope. The media were changed whenever the cell growth reached the confluence necessary for nutrient renewal. A quantity of 0.25% trypsin was added for detaching cells from the bottle walls. Cell cultures were negative for Mycoplasma sp. as assessed by a specific kit (Mycoplasma Stain Kit, Cat. MYC1, Sigma-Aldrich^®^, St Louis, MO, USA.

### 2.7. Cytotoxicity Activity

To evaluate the cytotoxicity of the biopigment, the Alamar blue test, based on resazurin [38], a fluorescent/colored indicator with redox properties, was performed 72 h after the exposure to the supernatant. Samples of crude pigment extract were diluted in sterile DMSO 10 mg mL^−1^ for extracts and 5 mg mL^−1^ DMSO for pure compounds and tested in concentrations that ranged from 0.19 to 50 µg mL^−1^. The cells were plated in 96-well plates (100 µL/well of a 0.7 × 105 cells mL^−1^ solution) and cultured for 24 h, and the tested substances dissolved in DMSO were added to each well and incubated for 72 h. Doxorubicin (doxorubicin hydrochloride, purity ≥ 95%, Laboratory IMA S.A.I.C., Buenos Aires, Argentina) was used as a positive control for cell toxicity. The negative control received the same amount of pure DMSO. A quantity of 20 µL of Alamar blue was added into each well 4 h before the end of the incubation period. Absorbances were then measured at 570 nm (reduced form) and 595 nm (oxidized form) [39]. Percentages of inhibition were calculated as a function of the log of the concentration to determine the IC_50_ using non-linear regression in the GraphPad Prism 5.0 software.

### 2.8. Chromatography Procedures

For thin layer chromatography (TLC), commercial plates with silica gel F254 as the stationary phase and dimensions of 20 cm × 20 cm × 0.20 mm were used for analytical TLC. High-performance liquid chromatography (HPLC) was performed on an Agilent G1311C-1260 quaternary pump coupled to a UV-Vis diode array (DAD), model G1315D-1260, using the semi-preparative column Phenomenex C-18 Gemini (4.6 × 250 mm, 5 µm) (Torrance, CA, USA) and HPLC-grade solvents.

For the chromatographic separation of compounds **1** and **2**, two different mobile phases were used for HPLC analysis during the elution gradient. Eluent A consisted of HPLC-grade water while eluent B consisted of methanol. The flow rate was 1 mL min^−1^. The total analysis time was 60 min, using the following gradient conditions: 0–50 min, 10–100% B; 50–55 min, 10% B (to clean the column), and 55–60 min 10% B (to equilibrate the column at the start conditions prior to each run).

The extract (349 mg) was chromatographed by column chromatography (H = 60 cm, 
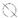
 = 4 cm) using silica gel 60 (70–230 mesh, MercK, Rahway, NJ, USA), starting with a mixture of EtOAc:n-Hex (9:1), gradually increasing the polarity up to 100% EtOAc, and finally increasing the MeOH percentage in the mixture up to a ratio of EtOAc:MeOH (1:1). A total of 240 fractions (5 mL each) were collected and, after analytical TLC, were grouped by similarity in eight fractions. Fraction 4 (14.1 mg) was revealed with Dragendorff’s reagent and showed the presence of alkaloids. Then, fraction 4 was submitted to high-performance liquid chromatography (HPLC). The compounds cyclo(L-Pro-L-Val) (21.8 min, 2.0 mg) and cyclo(L-Leu-L-Pro) (26.3 min, 3.0 mg) were isolated and identified by NMR spectra (Nuclear Magnetic Resonance) recovered on a Varian 400 MHz instrument (Santa Clara, CA, USA) using deuterated methanol (CD3OD) as solvent and tetramethylsilane (TMS) as the internal standard. All solvents used for the extraction and isolation procedures were of analytical grade.

### 2.9. Statistical Analysis

Statistical analysis was performed using the non-parametric Kruskal–Wallis test. The significance was set at 95% confidence interval. In addition, multiple comparisons were made to identify whether there was a statistical difference over time in each treatment.

## 3. Results

### 3.1. Fungal Molecular Identification

Two fungal molecular markers, LSU and ITS, were amplified using PCRs and subsequently sequenced using the Sanger methodology. The sequences were compared to other ITS and LSU fungal sequences deposited at the GenBank (deposit codes PQ 685987 and PQ 680080, respectively). The LSU sequence showed higher identities (ranging from 99.21 to 98.91%) with LSU sequences already deposited at the GenBank of *Pseudofusicoccum stromaticum*, *Pseudofusicoccum olivaceum*, *Pseudofusicoccum ardesiacum*, and *Pseudofusicoccum adansoniae* (Appendix A). When analyzing the ITS sequence of the fungal isolate, the same presented a higher identity (98.2 to 97.45%) with *Pseudofusicoccum adansoniae* (Appendix A) with low E-values.

The evolutionary history was inferred using the Maximum Likelihood method and the Jukes–Cantor model when considering the phylogenetic tree based on the LSU sequence. The tree with the highest log likelihood (−753.64) is shown in Figure 3. The initial trees for the heuristic search were obtained automatically by applying the Neighbor-Join and BioNJ algorithms to a matrix of pairwise distances estimated using the Jukes–Cantor model and then selecting the topology with the upper log likelihood value. The tree has been drawn to scale, with branch lengths measured by the number of substitutions per position. This analysis involved six nucleotide sequences. The codon positions included were the first + second + third + non-coding. There were a total of 505 positions in the final dataset. Regarding the ITS sequence, the evolutionary history was inferred using the Maximum Likelihood method and the three-parameter Tamura model. The tree with the highest log likelihood (−1443.33) is shown in Figure 3. The initial trees for the heuristic search were obtained automatically by applying the Neighbor-Join and BioNJ algorithms to a matrix of pairwise distances estimated using the three-parameter Tamura model and then selecting the topology with the upper log likelihood value. The tree has been drawn to scale, with branch lengths measured by the number of substitutions per position. This analysis involved six nucleotide sequences. The codon positions included were the first + second + third + non-coding. There were a total of 911 positions in the final dataset. In this way, it was possible to infer that the fungal isolate analyzed in this study belongs to the genus *Pseudofusicoccum*, *hereinafter* referred to as *Pseudofusicoccum* sp.

### 3.2. Effect of Light in Pigment Yield

The absorption spectra of the pigments under different light conditions and darkness revealed that the pigment composition varied significantly depending on the type of light and its specific conditions (lumens, exposure time, and whether the light was warm). Incubation under yellow light and total darkness resulted in the highest pigment production in the observed spectra (yellow, reddish-brown, and blue), followed by red, green, and blue light, while the lowest production occurred under white light (Figure 4).

At the 420 nm spectrum, which corresponds to maximum absorbance for yellow pigments, yellow light led to the highest pigment production (absorbance value of 1.66) whereas white light resulted in the lowest production (absorbance value of 0.23) (Figure 4A). A similar pattern was observed at the 520 nm spectrum, the peak for reddish-brown pigments, where yellow light yielded the highest pigment production (absorbance value of 3.56) and white light the lowest (absorbance value of 0.34) (Figure 4B). At the 620 nm spectrum, associated with blue pigments, the highest absorbance value was 0.46 under yellow light while the lowest was 0.06 under white light (Figure 4C).

These results may have been associated with a specific receptor for yellow light [29] that could enhance pigment production by the fungus. However, further studies are needed to confirm this hypothesis. A lower pigment production was observed in the spectrum corresponding to maximum absorbance for blue pigments (620 nm) under all conditions compared to the production observed in the 420 nm and 520 nm spectra. The maximum absorbance values were 0.46, 1.66, and 3.56, respectively (Figure 4).

### 3.3. Effect of Light on Growth

The effects of different wavelengths of light on biomass production were evaluated by measuring the dry biomass weight after 21 days of incubation (Figure 5). In submerged culture, there were variations in mycelial biomass weight over the 21-day period. The biomass produced in total darkness was the lowest (2.12 g) while the highest biomass was observed under yellow light (2.85 g), representing a 25.5% difference (Figure 5).

### 3.4. Cytotoxicity Assay

The cytotoxicity assays of the crude extract pigment produced by the *Pseudofusicoccum* sp. (Table 1) showed IC_50_ measurements for raw pigment against four cell lines. No cytotoxic effect was observed for any of the cell lines at the concentrations tested herein after 72 h of exposure since the IC_50_ was greater than 50 μg mL^−1^ for all cell strains, which was at least 100 times higher than that shown by the positive control doxorubicin.

### 3.5. Compound Identification Using NMR

The compounds *cyclo*(L-Pro-L-Val) **1** and *cyclo*(L-Leu-L-Pro) **2** (Figure 6) were identified in the pigment extract produced by the *Pseudofusicoccum* sp. using their NMR data compared with the literature [40,41].

### 3.6. NMR Analysis

The NMR analysis of compound **1** showed signals compatible with two doublet terminal methyls at 1.09 and 0.93 ppm that coupled to each other with a coupling constant of 6.9 Hz. A multiplet signal at 2.48 (m) ppm and a deshielded signal at 4.03 (m) ppm suggested the presence of a valine in the structure of compound **1**. Another deshielded signal at 4.20 ppm, neighboring an electronegative heteroatom together with a set of multiplet signals, suggested the structure of a proline in the structure. By comparing the experimentally obtained NMR data with the data obtained in the literature, compound **1** was identified as diketopiperazine cyclo(L-Pro-L-Val). Compound **2** showed the same pattern of signals observed for compound **1** with the addition of a multiplet signal at 2.06–1.84 (m) that integrated for two hydrogens, suggesting an additional methylene in the structure compatible with the structure of a leucine. By comparing the experimentally obtained NMR data with the data obtained in the literature, compound **2** was identified as diketopiperazine cyclo(L-Leu-L-Pro).

## 4. Discussion

This study presented significant results related, above all, to the identification and safety of the metabolites produced by the studied fungus. The data obtained corroborate those already reported in the literature but stand out for involving a fungus that had not yet been explored in the specific conditions of pigment production applied in this study. As examples, pigments produced by different fungi can be cited: the orange-brown pigment from endophytic *Aspergillus* sp. N11 [42], a red pigment (*Penicillium purpurogenum*) [43], red pigments (endophytic *M. ruber* SRZ112) [44], yellow dyes (*Arcopilus aureus*) [45], and a brown pigment (*Aspergillus ustus*) [46]. Pigments such as sclerothiorin (yellow), rubropunctamine (red), and bostrycoidin (red) are produced by different species of fungi [21]. Pereira et al. [9] isolated and identified pigment-producing endophytic fungi (*Aspergillus welwitschiae*, *A. sydowii*, *Curvularia* sp., *Diaporthe cerradensis*, *Hypoxylon investiens*, *Neoscytalidium* sp., and *Penicillium rubens*) from the Amazonian species *Fridericia chica*.

In 2006, fungi with pigmented and septate ascospores from the *Botryosphaeriaceae* family were studied through phylogenetic analysis, and some of these were classified into the *Pseudofusicoccum* genus [47]. In 2008, *P. adansoniae* was first described as an endophytic fungus associated with apparently healthy sapwoods and the barks of baobabs from Australia [48]. This genus is not well understood, but it has already been reported that extracts from these fungi have activity against phytopathogens [49]. While *Pseudofusicoccum* species have been described as having pigmented ascospores, little has been documented regarding pigment production by this specific genus. Our study represented a pioneering evaluation of this potential. Filamentous fungi can sense light and use it as a signal for physiological and morphological responses. Light plays a key role in regulating growth, pigment production, metabolism, and both asexual and sexual reproduction [27,50,51]. In a similar study by Velmurugan et al. [31], incubation in total darkness also resulted in increased pigment production, followed by the production of red, blue, unshielded white light, green, and yellow, in terms of the extracellular pigment yield across all isolates. These results align with those of Miyake et al. [30], who reported variations in the concentration of secondary metabolites (aminobutyric acid, red pigments, monacolin K, and citrinin) according to light wavelengths. However, this study did not evaluate light intensity, which could be an important variable for future analysis. These findings did not align with those of Palacio-Barrera, where biomass concentration was also higher under blue and green light. Zheng et al. [29] studied the influence of light on submerged cultures of *Inonotus obliquus* and found that the greatest increase in mycelial biomass occurred under red and blue light. This suggests that photoreceptors in *I. obliquus* can sense blue and red light, activating signal transduction pathways that favor mycelial growth under submerged culture conditions. Similarly, Babitha et al. [27] found that direct illumination favored growth while total darkness led to a reduction in biomass in *Monascus pupureus* culture.

Due to environmental stresses, such as exposure to light in its various spectra, many fungi biosynthesize secondary metabolites, which manifest as pigmentation [52]. These pigments can serve as adaptive responses to defense demands, forming protective barriers against the degradation of their mycelia by enzymes secreted by other microorganisms [53]. Additionally, their production may be a byproduct of the dormancy process [54].

The mechanisms regulating pigment production in fungi are diverse. Studies indicate that *quorum sensing*, which regulates phenotypes such as bioluminescence, also influences pigment synthesis.

Light plays a crucial role in regulating these processes, with different wavelengths influencing the production of distinct pigment types. Blue light (400–500 nm) is particularly associated with the regulation of genes involved in melanin and carotenoid synthesis, mediated by photoreceptors such as cryptochromes and LOV (Light, Oxygen, or Voltage) proteins. In *Aspergillus nidulans*, blue light stimulates the production of secondary pigments such as spiroquinones and other polyketides. Meanwhile, red light (620–750 nm), perceived by phytochrome-type photoreceptors, can also modulate pigment production, although its influence is generally less significant compared to blue light. On the other hand, UV radiation (10–400 nm), especially in UV-A frequencies (315–400 nm), induces the biosynthesis of photoprotective pigments such as melanin in *Cryptococcus neoformans*, protecting fungi against radiation-induced damage [55,56,57,58,59].

Among the different wavelengths, blue light and UV-A stand out as the primary modulators of pigment production in fungi, activating specific photoreceptors that regulate the genes involved in this process. However, the response varies depending on the fungal species and the type of pigment produced [52].

Lin and Xu [52] discuss, in their review, some biochemical mechanisms for the production of pigments such as carotenoids, melanins, and others in various fungal species.

In the case of *Pseudofusicoccum* sp., there is no description of the pigment production mechanisms in the literature, but examples from other well-studied fungi can be cited: *Blakeslea trispora* produces β-carotene through the cyclization of lycopene, a red intermediate in the β-carotene biosynthesis pathway, which is converted from isopentenyl pyrophosphate (IPP, C5) via the mevalonate (MVA) pathway, with acetyl-CoA as the precursor. *Fusarium *spp. primarily produces neurosporaxanthin, an apocarotenoid acid, derived from geranylgeranyl pyrophosphate (GGPP) through the activity of four enzymes encoded by the carRA, carB, carT, and carD genes. The initial steps in β-carotene biosynthesis involve the sequential addition of IPP units to form geranyl pyrophosphate (GPP, C10), farnesyl pyrophosphate (FPP, C15), and GGPP (C20).

Regarding melanin, the review highlights three main types produced by fungi: DOPA-melanin (immobilized in the cell wall), DHN-melanin, and pyomelanin. *C. neoformans* synthesizes DOPA-melanin through a series of oxidation–reduction reactions, starting with tyrosine or L-3,4-dihydroxyphenylalanine (L-dopa). *A. fumigatus* utilizes the DHN-melanin biosynthetic pathway, encoded by a gene cluster consisting of six genes: abr1, abr2, ayg1, arp1, arp2, and pksP/alb1.

Finally, *F. fujikuroi* produces bikaverin (6,11-dihydroxy-3,8-dimethoxy-1-methyl-benzo-xanthine-7,10,12-trione) through a polyketide biosynthetic pathway, requiring polyketide synthase (PKS) and using precursors such as acetyl-CoA and malonyl-CoA.

According to the Cytotoxic Drug Screening Program, extracts with IC_50_ > 30 ug mL^−1^ and pure compounds with IC_50_ > 4 µg mL^−1^ are considered of low toxicity and are seen as promising for the food industry [60,61]. The toxicity of some pigments produced by fungi has been previously evaluated against different types of human cells. Ethyl acetate extracts of *Aspergillus* sp. XJA6 and *A. terreus* XJA8 have been tested against human Hela and HT-29 (colon cancer) cells, showing an IC_50_ of 9.99 ± 0.80 µg mL^−1^ and 5.73 ± 0.60 µg mL^−1^, respectively [62]. A pigment extracted from *F. chlamydosporum* mycelia has exhibited moderate levels of cytotoxicity against human cancer cell lines, although it has exhibited less cytotoxicity against CHOK 1 cells [63]. The endophytic fungus *Aspergillus tamarii*, isolated from *Ficus carica* roots, was used in a study to produce secondary metabolites under submerged fermentation. The obtained extract showed cytotoxic activity against human cancer cells such as MCF-7 and A549 [25]. Compounds isolated from the endophytic fungus *Aspergillus* sp. showed a cytotoxic effect against the A549 cell line in a study [26]. The results obtained in this study suggest that the crude extract of pigment produced by *Pseudofusicoccum adansoniae* did not show toxicity at the concentrations tested, which makes it a promising product to be used as a dye in the food industry; however, additional studies, including mycotoxins, need to be further explored so a more inclusive toxicity panel of the pigment can be constructed [64]. Cyclic dipeptides (diketopiperazines) can be produced by different sources and have different biological activities described in the literature [65,66].

Diketopiperazines have been detected in a variety of natural products, as well as in processed foods, beverages, and food and beverage ingredients [16]. The cyclic dipeptides are generally reported as weakly bitter and slightly astringent. Chen and colleagues [67] discovered that compounds **1** and **2** (Figure 5) were found to occur in cooked beef and the compound **2** concentration of 20.6 ppm in stewed beef was higher than its taste threshold value (about 10 ppm). Chen and colleagues showed the presence of these compounds in chicken essence [68] and observed their presence in beer [16], in roasted coffee brews [69], in roasted malt [70], and in roasted cocoa nibs [71]. Compounds **1** and **2** were isolated from endophytic *Streptomyces* sp. and exhibited activity against methicillin-resistant *Staphylococcus aureus* and *Enterococcus raffinosus*, with low toxicity against human hepatoma HepaRG cells [72], and compound **2**, isolated from *Streptomyces* sp., was shown to be active against twelve vancomycin-resistant enterococci strains, including *Enterococcus faecium* (vanA, vanB) and *E. faecalis* (vanA, vanB), and was effective against three leukemic cell lines at concentrations below 100 mg ml^−1^ [73].

*Cyclo*(L-Pro-L-Val) (compound **1**) showed antifungal activity against *Ganoderma boninense* and *Candida albicans* [74] and compound **2**, isolated from *Streptomyces* sp., exhibited activity against *Candida albicans*, *Mucor ramannianus*, *Rhizoctonia solani*, *Aspergilus fumigatus*, *Glomerella cingulata*, *Trichophton mentagrophytes*, and *Trichophyton rubrum*; the order of MIC values was 50.0, 12.5, 5.0, 50.0, 25.0, 5.5 µg ml^−1^. Specifically, compound **2** was one of the most effective compounds against *Pyricularia oryzae,* with an MIC value of 2.5 µg ml^−1^, thus indicating its potential antifungal agent [75]. Both compounds have been described as quorum-sensing molecules isolated from *Burkholderia cepacian* [76], and compound **2**, produced by *Pseudomonas putida,* is able to activate the quorum-sensing biosensors of *Chromobacterium violaceum* and *Agrobacterium tumefaciens* [77]. In a study, *Achromobacter xylosoxidans* produced compound **1** and **2**, which inhibited aflatoxin production by *Aspergillus parasiticus* [78], and compound **1** isolated from *Stenotrophomonas* spp. inhibited aflatoxin production in *Aspergillus parasiticus* and *Aspergillus flavus* in liquid medium at concentrations of several hundred µM without affecting fungal growth [79].

*Cyclo*(L-Pro-L-Val) and *cyclo*(L-Leu-L-Pro) (compounds **1** and **2**) were isolated from the deep-sea bacterium *Streptomyces fungicidicus* and were tested for antilarval activity using the barnacle *Balanus amphitrite*. The results suggested that these compounds could be used as novel antifoulants [80], and these same compounds were isolated from *Pseudomonas rhizosphaerae* obtained from deep-sea sediments and compound **1** showed antilarval effects on the larval settlement of the barnacle *Balanus amphitrite* and bryozoan *Bugula neritina* [81]. In a study, *Pseudomonas fluorescens*, isolated from the pine wood nematode, *Bursaphelenchus xylophilus*, produced compound **1**, which was toxic to both suspension cells and seedlings of *Pinus thunbergii,* indicating that it could be responsible for the disease caused by *B. xylophilus* [82]. In other studies, compound **1** isolated from *Aspergillus* sp. was a specific *β*-glucosidase inhibitor [83], and compound **1** isolated from *Streptomyces* sp. showed weak cytotoxicity against SV40-transformed cells [84].

In summary, the findings of this study hold significant industrial relevance, highlighting the characterization of a pigment extract that exhibits no cytotoxicity and contains compounds with well-recognized and promising activities for industrial applications.

## 5. Conclusions

The originality of this study lay in the molecular identification of a pigment-producing fungus, *Pseudofusicoccum* sp. The growth of this fungus was not affected by exposure to different light wavelengths. However, pigment production varied depending on the color of the light used during incubation. The extract produced by the fungus showed no cytotoxicity, and compounds with beneficial biological activities were identified. These preliminary results highlight the potential application of the obtained extract, suggesting the need for future studies, particularly focusing on the identification of the other compounds present and the evaluation of the biological activities of the extract. Subsequent work will investigate the physicochemical stability of the produced pigment.

## Figures and Tables

**Figure 1 microorganisms-13-00277-f001:**
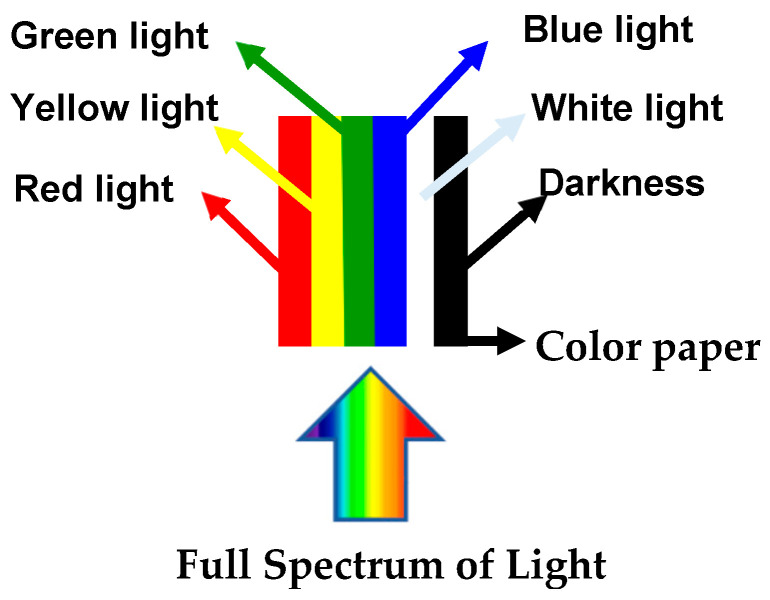
Principle used to conduct the experiment. The flasks were covered in colored glass papers of blue (492–455 nm), green (577–492 nm), yellow (597–577 nm), or red (780–622 nm) and placed at an illuminated light source inside the incubator installed with light (G-light, 18 W) and darkness.

**Figure 2 microorganisms-13-00277-f002:**
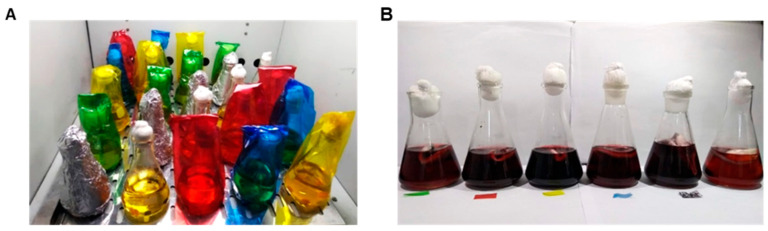
(**A**) Experimental setup to study the effects of different light sources on growth and extracellular pigment production for *Pseudofusicoccum* sp. (**B**) Submerged culture after 21 days of incubation.

**Figure 3 microorganisms-13-00277-f003:**
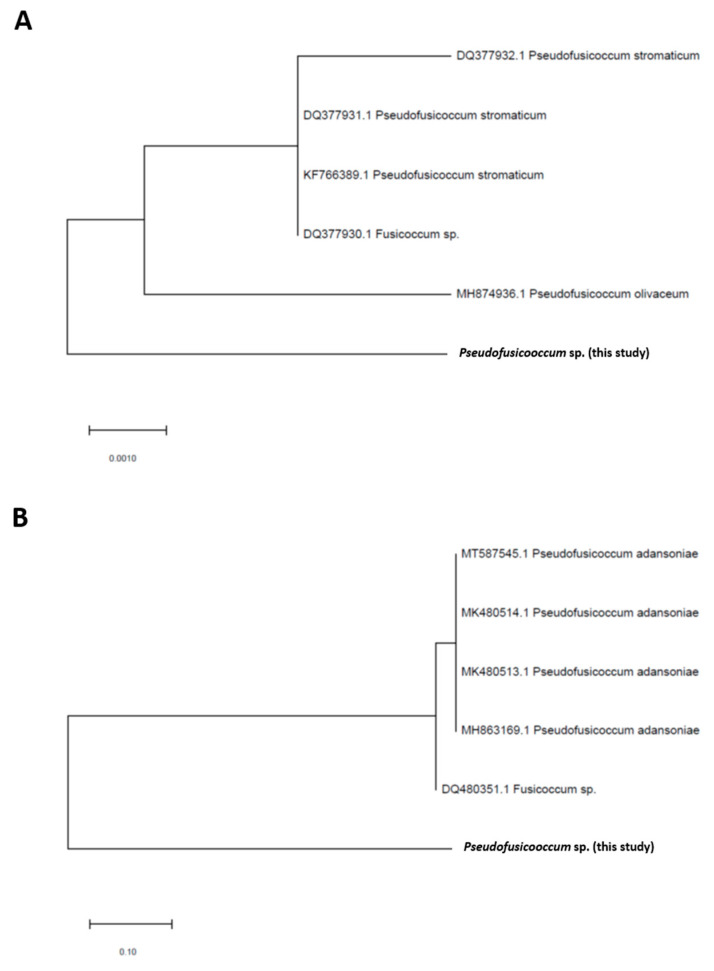
Phylogenetic trees built with the (**A**) LSU and (**B**) ITS sequences of the isolated fungus studied herein. Fragments of both genes were amplified using PCR, sequenced, and aligned, and the best phylogenetic model was selected using the MEGA software (version 11).

**Figure 4 microorganisms-13-00277-f004:**
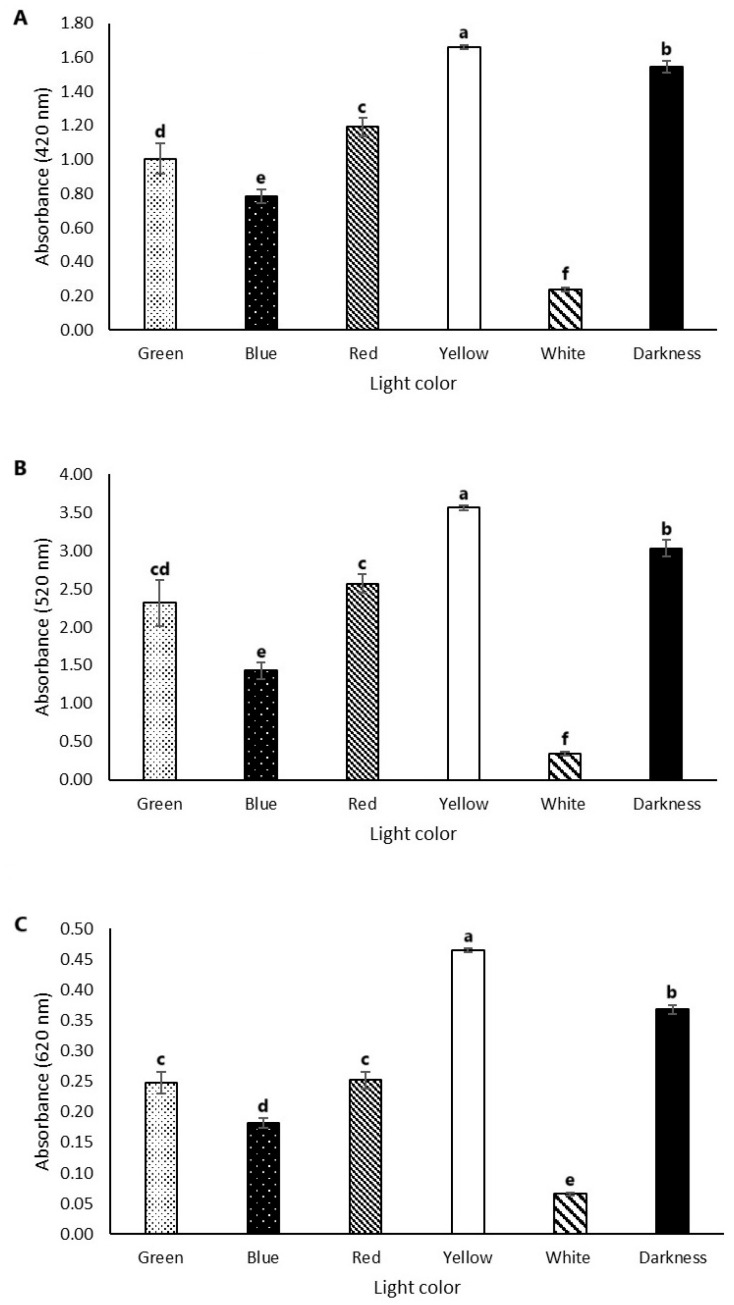
Effects of different colors of light on pigment production. The samples were submitted to different colors (yellow, blue, red, green, white, and darkness) and analyzed by spectrophotometry (Multiskan FC—Thermo Scientific); wavelengths: (**A**) 420 nm, (**B**) 520 nm and (**C**) 620 nm. The error bars represent a 95% confidence limit for the measurements. Different letters indicate significant differences by Tukey’s test (*p* ≤ 0.05; n = 4).

**Figure 5 microorganisms-13-00277-f005:**
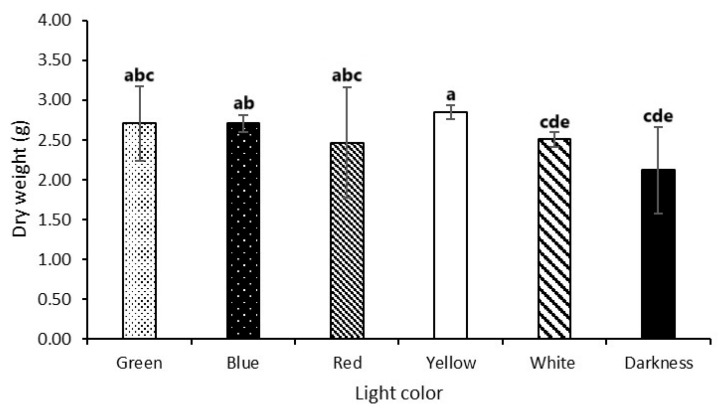
Effect of light on growth. Samples submitted to different lights (yellow, blue, red, green, white, and darkness) containing mycelia were filtered and the separated mycelia were washed twice with deionized water, followed by drying at 105 °C (37,815 K) for 12–15 h, and weighed to yield the biomass. The error bars represent 95% confidence limits for the measurements. Different letters indicate significant differences by Tukey’s test (*p* ≤ 0.05; n = 4).

**Figure 6 microorganisms-13-00277-f006:**
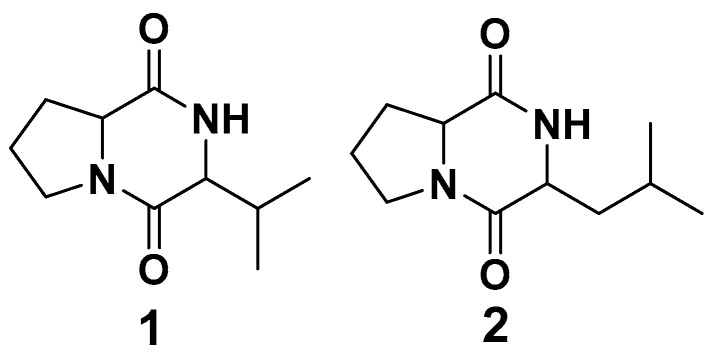
Chemical structures of compounds **1** and **2**.

**Table 1 microorganisms-13-00277-t001:** IC_50_ of the *Pseudofusicoccum* sp. crude pigment extract against four human cell lineages. The cells include in this experiment were the following: HepG2—human hepatocellular carcinoma, SCC4—oral human squamous carcinoma, BJ—human foreskin fibroblast, and MRC-5—human lung fibroblast), all of them obtained from the American Type Culture Collection (ATCC). The cells were incubated with the supernatant containing the pigment for 72 h. Doxorubicin was used as a positive control. The results express the means of three independent experiments.

Samples	IC_50_ (μg mL^−1^)			
	HepG2	SCC4	BJ	MRC-5
*Pseudofusicoccum* sp. crude pigment extract	>50	>50	>50	>50
Doxorubicin	0.05	0.13	4.84	2.16
	0.03–0.11	0.10–0.18	2.94–0.86	0.75–6.17

## Data Availability

The original contributions presented in the study are included in the article/Appendix A, further inquiries can be directed to the corresponding authors.

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
