# Peer review of "Pigment Production by *Pseudofusicoccum* sp.: Extract Production, Cytotoxicity Activity, and Diketopiperazines Identified"

_microorganisms, 2025, doi:10.3390/microorganisms13020277_

Round 1

Reviewer 1 Report

Comments and Suggestions for Authors

The submitted manuscript is about an interesting work on fungus pigment production with important application as a natural pigment. In particular, pigment synthesis influenced by light and separation pigment are the pillars of this wok. However, I have few concerns about it.

1. As a general comment, I would like to say that the manuscript can be reviewed by a native English-speaking colleague so that he can further improve the English in introduction and disscution sections .

2. There are some typographical errors and many problems with the format should be corrected in text.

(1)   line 96 “IT4/ITS5 rimers” should be “ITS4/ITS5 primers”.

(2)   The symbol number should be superscripted and and subscripted. Line 145/147/190 and so on.please check the text.

(3)   line 249-252 should be placed after Fig. 4. The decimal point of the vertical axis of A/B/C should be "." Instead of ", ".

(4)   line 265 “sp.”should be black; line268“ug “should be “μg”. IC50 requires IC50 in text and table1.

3. line206-225, the ITS and LSU sequences of strains should be submitted to GenBank for deposition, and obtained GenBank ID.Strains identification described too cumbersome by phylogenetic trees, and give a brief description through reference literature.

4. The discussion section is not highly relevant to the research objectives and significance of the text. The discussion section indicates that many fungi can produce pigments, but there are few reports on the pigment production by the Pseudofusicoccum genus, making this study a pioneering evaluation of such potential. However, it remains to be determined whether the isolated compounds, cyclo(L-Pro-L-Val) 1 and cyclo(L-Leu-L-Pro) 2, are pigment components produced by Pseudofusicoccum sp.. Additionally, the discussion highlights that these cyclic dipeptides have antimicrobial and antilarval  properties, but no established connection with pigment applications.

Comments on the Quality of English Language

 the manuscript can be reviewed by a native English-speaking to further improve the English 

Author Response

Thank you very much for your comments. The questions or suggestions you made have been answered.

Reviwer 1

The submitted manuscript is about an interesting work on fungus pigment production with important application as a natural pigment. In particular, pigment synthesis influenced by light and separation pigment are the pillars of this wok. However, I have few concerns about it.

  1. As a general comment, I would like to say that the manuscript can be reviewed by a native English-speaking colleague so that he can further improve the English in introduction and discussion sections .
  2. There are some typographical errors and many problems with the format should be corrected in text.

(1)   line 96 “IT4/ITS5 rimers” should be “ITS4/ITS5 primers”.

Thank you for your observation. The correction has been made.

(2)   The symbol number should be superscripted and and subscripted. Line 145/147/190 and so on.please check the text.

Thank you for your observation. The correction has been made.

(3)   line 249-252 should be placed after Fig. 4. The decimal point of the vertical axis of A/B/C should be "." Instead of ", ".

Thank you for your observation. The correction has been made.

(4)   line 265 “sp.”should be black; line268“ug “should be “μg”. IC50 requires IC50 in text and table1.

Thank you for your observation. The correction has been made.

  1. line206-225, the ITS and LSU sequences of strains should be submitted to GenBank for deposition, and obtained GenBank ID.Strains identification described too cumbersome by phylogenetic trees, and give a brief description through reference literature.

Thank you for your observation. The sequences are being deposited in genbank. Waiting for the deposit code to be generated. The identifier will be inserted in the final version of the article.

  1. The discussion section is not highly relevant to the research objectives and significance of the text. The discussion section indicates that many fungi can produce pigments, but there are few reports on the pigment production by the Pseudofusicoccumgenus, making this study a pioneering evaluation of such potential. However, it remains to be determined whether the isolated compounds, cyclo(L-Pro-L-Val) and cyclo(L-Leu-L-Pro) 2, are pigment components produced byPseudofusicoccum sp.. Additionally, the discussion highlights that these cyclic dipeptides have antimicrobial and antilarval  properties, but no established connection with pigment applications.

Thank you for your observation.

Claims about antimicrobial properties are directly related to food applications, considering that if the extract contains both pigments of industrial interest and compounds with antimicrobial activity, this will have significant relevance for the safety and microbiological stability of the products in which it is applied.

The characteristics of the compounds cyclo(L-Pro-L-Val) 1 and cyclo(L-Leu-L-Pro) 2 will be detailed and discussed in future studies. In this work, the focus is to identify the fungus, determine the best light conditions for the production of the extract, evaluate its cytotoxicity and characterize some of the first compounds isolated.

Reviewer 2 Report

Comments and Suggestions for Authors

Dear authors, 

thank you for providing the paper about fungus and cytotoxity. 

Overall opinion

The research idea and hypothesis are clear, the research design is appropriate and the methods are well described. 

Minor changes to be made

Introduction: 

the introduction section should be revised in the sense that general topics should be set first about the global perspective (lines 56 to 61)

then 

highlight the problem of artificial colours and damage to the environment and human health

then 

show the advances and possibilities of pigments from fungus ( lines 5o to 65) 

then

problems of toxity

then 

specific fungues colours in  lines 35 to 44

In materials and methods

explain why did You chose the M.S. tree ? (line 85)

is it important for brazil 

or the industry of because of colour ?

Discussion 

the papers about fungus , lines297 to 310 should be part of Introduction 

also lines332 to 337 to be put in introduction

the toxicity problem in lines 338 onwards can be put as part of introduction as separate sub section

Insteda of current discussion focus on Your finding and the possibilities of M.S. leaves and fungus use in Brazil or wider. 

kind regards, 

the reviewer. 

Reviewer 3 Report

Comments and Suggestions for Authors

Dear Authors,

I had the opportunity to review your article entitled “Pigment Production by Pseudofusicoccum sp.: Extract Production, Cytotoxicity Activity, and Diketopiperazines Identified.” I found the results and approach employed to be interesting; however, the article requires copyediting in terms of language, organization of ideas, and presentation. Furthermore, the discussion is too condensed and does not clearly and effectively convey the meaning of the results. Although the methodology is coherent and the results are promising, I suggest the authors make the necessary corrections. In its current state, the article needs significant improvements, and I cannot consider it for publication. If the authors address these issues and enhance the content and presentation, I would be happy to evaluate it again.

I recommend that the authors revise the English syntax, as some excerpts need improvement.

Specific Comments:

Page 1:

- Correct: “Correspondence: marcelo.umsza@ufba.br” (remove the redundancy)

- Abstract: I suggest presenting the abstract in a more coherent manner. Instead of first stating what the article aims to do and then what was done, consider starting by describing the results. This way, the abstract can focus more on the findings and their industrial significance.

- Remove redundancy in the phrase “Biotechnology in general, as well as biotechnological processes.”

- In the phrase “new and/or hyperproducing strains,” note that a hyperproducing strain is inherently a new strain.

- Review the length of paragraphs in the introduction. Avoid paragraphs that are too short or too long (see the first and second paragraphs as an example).

- “Filamentous fungi are the most used to produce dyes [6,7].” Consider rephrasing this to: “Filamentous fungi are among the most commonly used microorganisms for dye production.”

Page 2:

- The conclusion of the introduction would benefit from explicitly stating the hypothesis of this work and what was done to investigate it. Also, highlight the novelty of the study.

- Verify the accuracy of the coordinates “(12°56’59” S and 38°20’25” W).”

Page 3:

- I commend the team for their creativity in setting up the experiment. However, additional details are needed to ensure precise reproducibility:

- Full specifications of the 18W light source: Is it LED? What is the color temperature and lumen output?

- Specify the placement inside the photoincubator and the distance from each sample. Each flask should receive an equal amount of lumens.

- Did the authors measure the lumen output after the white light passed through the filters? If so, this information should be included in the methodology.

- I suggest including drawn schematics to support the experimental setup instead of just using the process photograph.

Page 6:

- L232: Clarify “light and its conditions.”

Page 7:

- Fig. 4: Clarify the meaning of “The samples were submitted to different colors clear.”

- Ensure the use of “.” instead of “,” when appropriate in the figures.

- Standardize the resolution of figures. Figure 4C, for instance, has a lower resolution.

- Provide statistical analysis to indicate the significance between each group in the column graphs.

Page 8:

- Figure 5: Why are the standard deviation bars higher for the green, blue, and white groups?

- Review Line 265 for clarity.

- Standardize the font type and size in Table 1 and throughout the manuscript.

- L281: Remove the “-.” Revise the manuscript for small formatting errors.

Page 9:

- L297-305: Ensure the paragraph clearly articulates the value of your work in relation to existing literature.

- L317: Verify “n.”

- In the discussion section, as the results are divided, I suggest dividing the discussion into sections corresponding to each result. This will help organize the ideas more clearly. Also, provide supported explanations for each aspect of the results and cite relevant literature where necessary. Currently, the discussion reads more like an introduction or a section of a review article.

  Additionally, consider addressing the following:

  - What are the possible mechanisms involved in light-assisted bioprocessing with fungi?

  - According to existing literature, do similar genera have receptors for specific wavelengths? Which wavelengths may act as stressors, with pigment production as a protective response?

  - Which wavelengths may activate enzymes involved in biopigment production?

  - What is the industrial significance of biopigment production using the discoveries reported here for this strain in a real-world scenario?

- The third paragraph of the discussion is too long and difficult to follow. I strongly recommend breaking it into smaller sections for easier readability. Since the journal allows a combined results and discussion section, I believe this would be the best strategy to improve clarity and readability.

- Conclusion: The conclusion needs improvement. What insights can be drawn without repeating the results and discussion? What is the significance of your findings? What are the next steps for this research?

Round 2

Reviewer 1 Report

Comments and Suggestions for Authors

The article has been revised in accordance with the review comments and generally meets the requirements

Reviewer 3 Report

Comments and Suggestions for Authors

Dear Authors,  

I have had the opportunity to review the revised version of your manuscript titled “Pigment production by Pseudofusicoccum sp.: Extract production, cytotoxicity activity, and diketopiperazines identified.” However, the manuscript requires further revision, as it currently does not meet the minimum quality standards for publication.  

Comments:  

Abstract: The abstract must be significantly improved. Begin with a clear and concise introduction to the topic, followed by the specific results and their significance. In its current state, the abstract does not meet the necessary quality for publication.  

Tracked Changes: It is difficult to follow the changes made in the revised version due to the use of yellow highlights and red text. Do these indicate different types of edits? The authors must use the “Track Changes” function to document all modifications, including deletions, so the referee can review all updates accurately.  

Hypothesis: Revisit the comments regarding the hypothesis at the end of the introduction. A clear and well-defined hypothesis is missing.  

Light Specifications: Specify the color of the light used in the experiments in terms of temperature (measured in Kelvin).  

Graph Formatting: Ensure uniform formatting of the Y-axes across all graphs. For example, use consistent decimal places (e.g., 4.00, 3.50) throughout the manuscript.  

Figure 5: Check for outlier values used to prepare Figure 5. Additionally, clearly state in the text the statistical differences between the groups.  

Discussion: Enhance the discussion section by clearly articulating the value of your work in relation to existing literature. For instance, the excerpt “This work, as well as many others already published, show the production…” fails to highlight the significance of your findings and may even detract from it.  

Mechanisms of Light-Assisted Bioprocessing: Incorporate a detailed discussion of the mechanisms involved in light-assisted bioprocessing with fungi. Provide specific insights into the mechanisms relevant to this fungal genus, as the current discussion is too general.  

Formatting and Editing: The manuscript contains several formatting and editing issues. For example, bold symbols are used where italics should be applied. Carefully review the manuscript to ensure a professional and consistent presentation.  

Content Incorporation: Address all questions and suggestions within the manuscript itself, rather than solely in the response letter to the referee. Ensure that all discussions and clarifications are integrated into the manuscript text.

Industrial Significance: Clearly state the industrial significance of biopigment production using the findings reported for this strain. The referee seeks to understand how these discoveries could influence or improve the production of microbial pigments in real-world industrial scenarios.  

Once the authors carefully revise the manuscript addressing the comments above, I will be available to review it again.

Author Response

Dear reviewer

We are very grateful for your consideration and suggestions. All points have been corrected and/or explained.

Comments:  

- Tracked Changes: It is difficult to follow the changes made in the revised version due to the use of yellow highlights and red text. Do these indicate different types of edits? The authors must use the “Track Changes” function to document all modifications, including deletions, so the referee can review all updates accurately.  

Answer: The journal requests that changes be marked in yellow in the text, showing where the modifications pointed out by the reviewers were made. In this version of the article, the modifications or changes requested are from a single reviewer, with the exception of the inclusion of the deposit number of the sequences in Genbank.

- Abstract: The abstract must be significantly improved. Begin with a clear and concise introduction to the topic, followed by the specific results and their significance. In its current state, the abstract does not meet the necessary quality for publication.  

Answer: Some adjustments were made to the abstract, as requested.

Filamentous fungi are among the most commonly used microorganisms for producing various metabolites, including dyes. Ensuring the safety of products derived from microorganisms is always essential. In this study, the isolated fungus was identified as Pseudofusicoccum sp., a producer of burgundy pigment through submerged fermentation. The fungus exhibited enhanced growth and pigment production under yellow light. The extract obtained showed no cytotoxicity in the tested cell lines (HepG2, SCC4, BJ, and MRC-5). Among the compounds isolated and identified through NMR analysis, cyclo(L-Pro-L-Val) and cyclo(L-Leu-L-Pro) (diketopiperazines) were previously reported in foods and are known to be produced by various organisms, with several beneficial biological activities. This identified fungus represents a promising source of biopigments, with a crude extract that is non-cytotoxic. Additionally, the isolated compounds exhibit significant biological properties, such as antibacterial, antifungal, and antioxidant activities, highlighting their potential as natural pigments for use in food products.

- Hypothesis: Revisit the comments regarding the hypothesis at the end of the introduction. A clear and well-defined hypothesis is missing. 

Answer: The hypothesis was inserted at the end of the introduction.

Considering that fungi produce various types of metabolites, particularly under stress conditions caused by process variables, this study aimed to identify the isolated fungus, evaluate the influence of light type on pigment production, analyze cytotoxicity, and identify some compounds present in the obtained extract. The main contribution of this research is to provide scientific evidence of the extract's potential and safety, focusing on cytotoxicity assessment and its potential future applications.

- Light Specifications: Specify the color of the light used in the experiments in terms of temperature (measured in Kelvin).

Answer:  As explained in the material and methods, and in Figure 2, the vials were covered by colored glass paper, so no lights of different colors were used.

“The flasks were covered in colored glass papers of blue (492–455 nm), green (577– 492 nm), yellow (597–577 nm) or red (780–622 nm) and placed under an 18W light source (LED Cold light, 1350 lumens - with a light source centralized) inside an incubator and kept at 28 ± 2°C for 21 days”.

  • Kelvin temperatures were entered in parentheses

- Graph Formatting: Ensure uniform formatting of the Y-axes across all graphs. For example, use consistent decimal places (e.g., 4.00, 3.50) throughout the manuscript.  

Answer: Thanks for the note. Those details have been corrected.

- Figure 5: Check for outlier values used to prepare Figure 5. Additionally, clearly state in the text the statistical differences between the groups.  

Discussion: Enhance the discussion section by clearly articulating the value of your work in relation to existing literature. For instance, the excerpt “This work, as well as many others already published, show the production…” fails to highlight the significance of your findings and may even detract from it.  

Answer: Thank you for your observation. The text has been revised and discussed again.

This study presents significant results related, above all, to the identification and safety of the metabolites produced by the studied fungus. The data obtained corroborate those already reported in the literature but stand out for involving a fungus that has not yet been explored in the specific conditions of pigment production applied in this study.

Mechanisms of Light-Assisted Bioprocessing: Incorporate a detailed discussion of the mechanisms involved in light-assisted bioprocessing with fungi. Provide specific insights into the mechanisms relevant to this fungal genus, as the current discussion is too general.  

Due to environmental stress, such as exposure to light in its various spectra, many fungi biosynthesize secondary metabolites, which manifest as pigmentation (Li and Xu, 2020). These pigments can serve as adaptive responses to defense demands, forming protective barriers against the degradation of their mycelia by enzymes secreted by other microorganisms [51]. Additionally, their production may be a byproduct of the dormancy process [52]. 

The mechanisms regulating pigment production in fungi are diverse. Studies indicate that quorum sensing, which regulates phenotypes such as bioluminescence, also influences pigment synthesis.

Light plays a crucial role in regulating these processes, with different wavelengths influencing the production of distinct pigment types. Blue light (400–500 nm) is particularly associated with the regulation of genes involved in melanin and carotenoid synthesis, mediated by photoreceptors such as cryptochromes and LOV (Light, Oxygen, or Voltage) proteins. In Aspergillus nidulans, blue light stimulates the production of secondary pigments, such as spiroquinones and other polyketides. Meanwhile, red light (620–750 nm), perceived by phytochrome-type photoreceptors, can also modulate pigment production, although its influence is generally less significant compared to blue light. On the other hand, UV radiation (10–400 nm), especially UV-A frequencies (315–400 nm), induces the biosynthesis of photoprotective pigments such as melanin in Cryptococcus neoformans, protecting fungi against radiation-induced damage [55–59]. 

Among the different wavelengths, blue light and UV-A stand out as the primary modulators of pigment production in fungi, activating specific photoreceptors that regulate the genes involved in this process. However, the response varies depending on the fungal species and the type of pigment produced.

 Lin and Xu, (2020), discuss in their review some biochemical mechanisms for the production of pigments, such as carotenoids and melanins, in various fungal species.

In the case of Pseudofusicoccum sp., there is no description of the pigment production mechanisms in the literature, but examples from other well-studied fungi can be cited: Blakeslea trispora produces β-carotene through the cyclization of lycopene, a red intermediate in the β-carotene biosynthesis pathway, which is converted from isopentenyl pyrophosphate (IPP, C5) via the mevalonate (MVA) pathway, with acetyl-CoA as the precursor. Fusarium spp. primarily produces neurosporaxanthin, an apocarotenoid acid, derived from geranylgeranyl pyrophosphate (GGPP) through the activity of four enzymes encoded by the carRA, carB, carT, and carD genes. The initial steps in β-carotene biosynthesis involve the sequential addition of IPP units to form geranyl pyrophosphate (GPP, C10), farnesyl pyrophosphate (FPP, C15), and GGPP (C20).

Regarding melanin, the review highlights three main types produced by fungi: DOPA-melanin (immobilized in the cell wall), DHN-melanin, and pyomelanin. C. neoformans synthesizes DOPA-melanin through a series of oxidation-reduction reactions, starting with tyrosine or L-3,4-dihydroxyphenylalanine (L-dopa). A. fumigatus utilizes the DHN-melanin biosynthetic pathway, encoded by a gene cluster consisting of six genes: abr1, abr2, ayg1, arp1, arp2, and pksP/alb1.

Finally, F. fujikuroi produces bikaverin (6,11-dihydroxy-3,8-dimethoxy-1-methyl-benzo-xanthine-7,10,12-trione) through a polyketide biosynthetic pathway, requiring polyketide synthase (PKS) and using precursors such as acetyl-CoA and malonyl-CoA.

- Formatting and Editing: The manuscript contains several formatting and editing issues. For example, bold symbols are used where italics should be applied. Carefully review the manuscript to ensure a professional and consistent presentation.  

Answer: Thank you for your observation. A check was performed to minimize this type of error.

- Content Incorporation: Address all questions and suggestions within the manuscript itself, rather than solely in the response letter to the referee. Ensure that all discussions and clarifications are integrated into the manuscript text.

Answer: Thank you for your suggestions. All changes have been highlighted and incorporated into the text.

- Industrial Significance: Clearly state the industrial significance of biopigment production using the findings reported for this strain. The referee seeks to understand how these discoveries could influence or improve the production of microbial pigments in real-world industrial scenarios.  

Answer: Thank you for the suggestion. At the end of the article, a paragraph was added referring to the industrial importance of the results found in this work.

In summary, the findings of this study hold significant industrial relevance, highlighting the characterization of a pigment extract that exhibits no cytotoxicity and contains compounds with well-recognized and promising activities for industrial applications.

Round 3

Reviewer 3 Report

Comments and Suggestions for Authors

Dear authors,

I have reviewed your revised manuscript and am satisfied with the improvements and corrections made